# Knowledge Discovery in COVID-19 Research Literature

**Alejandro Piad-Morffis**[1], **Suilan Estevez-Velarde**[1], **Ernesto L. Estevanell-Valladares**[1],
**Yoan Gutiérrez**[2,3], **Andrés Montoyo**[2,3], **Rafael Muñoz**[2,3], and **Yudivián Almeida-Cruz**[1]

[1]School of Math and Computer Science, University of Havana, Cuba
{apiad,sestevez,estevanell,yudy}@matcom.uh.cu
[2]University Institute for Computing Research (IUII), University of Alicante, Spain
[3]Department of Languages and Computing Systems, University of Alicante, Spain
{ygutierrez,montoyo,rafael}@dlsi.ua.es

## Abstract

This paper presents the preliminary results of an ongoing project that analyzes the growing body of scientific research published around the COVID-19 pandemic. In this research, a general-purpose semantic model is used to double annotate a batch of 500 sentences that were manually selected from the CORD-19 corpus. Afterwards, a baseline text-mining pipeline is designed and evaluated via a large batch of 100, 959 sentences. We present a qualitative analysis of the most interesting facts automatically extracted and highlight possible future lines of development. The preliminary results show that general-purpose semantic models are a useful tool for discovering fine-grained knowledge in large corpora of scientific documents.

## 1 Introduction

The COVID-19 pandemic that affected almost every country at the beginning of 2020 has provoked a massive increase in scientific papers related to biomedical sciences (Velavan and Meyer, 2020). The scientific community's effort to fight the spread of the virus is evident in the record number of research papers about COVID-19 that have been submitted to conferences, journals and preprint services. Several academic publishers joined the effort by providing free access to research in related areas that could be useful to scientists and academics. The amount of information produced during this period greatly surpassed the ability of human researchers to stay up-to-date, which in turn spawned an increased interest in the application of computational techniques to automatically organize, normalize and link the existing information.

A recent initiative is the *COVID-19 Open Research Dataset* (CORD-19), published by the Allen Institute for AI (Lo et al., 2020), which makes available a large corpus of scientific papers on COVID-19 and related topics. At the moment of writing, it contains 76, 674 scientific articles. The corpus has been used as part of a Kaggle challenge[1] , which focused mainly on unsupervised tasks related to organizing and categorizing the different aspects of the whole COVID-19 situation. Based on these resources, computational tools, e.g., SciSight (Hope et al., 2020), SciFact (Wadden et al., 2020), and similar (Bras et al., 2020), have been created to enable the interactive visualization and exploration of the scientific literature and the discovery of connections between the available methods, symptoms, interventions, etc.

Unsupervised approaches are a natural strategy for dealing with large, unlabeled corpora, while supervised approaches have the caveat of requiring training examples to be manually annotated, but can provide precise answers to specific questions given enough supervised data. For example, identifying domain-specific entities such as symptoms, medication and treatments, and semantic relations between them. Learning to recognize this type of information in natural language, even academic language, is a challenging task, given the large number of varieties in which the same semantic fact can be stated. In this context, different annotation models have been designed to capture the semantic meaning in different domains and levels of discourse. Token-level annotation models, such as AMR (Banarescu et al., 2013), capture fine-grained semantic relations between elements in a natural language sentence, independently of domain, which means that only general-purpose relations can be recognized. In contrast, domain-specific annotation models can capture more detailed relations, such as

---

[1]https://www.kaggle.com/allen-institute-for-ai/CORD-19-research-challenge

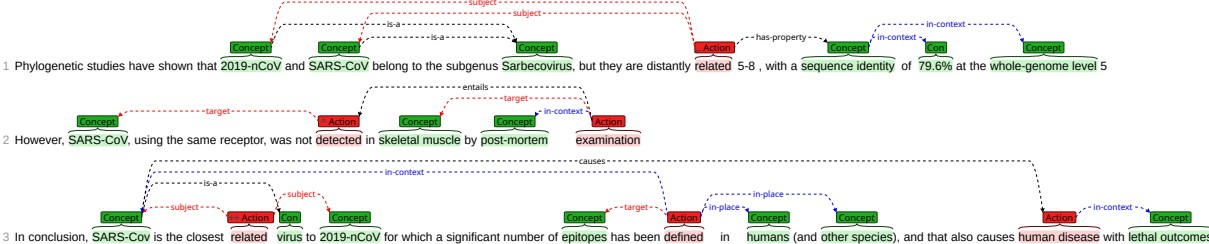

Figure 1: Example annotation of three sentences taken from the CORD-ANN corpus (batch 58). The annotation shows the most relevant entities and relations defined in the SAT+R annotation model.

drug-drug interactions (Herrero-Zazo et al., 2013) or adverse reactions (Karimi et al., 2015), but are less useful in cross-domain scenarios or in slightly different tasks.

The SAT+R (*Subject-Action-Target + Relations*) annotation model (Piad-Morffis et al., 2019a) attempts to achieve a middle ground between domain independence and specificity. It has been developed as a general-purpose model but evaluated in different health-related scenarios (Piad-Morffis et al., 2019c), specifically in Medline articles in Spanish language. This model proposes two different classes of semantic relations, teleological and ontological, and 4 classes of entities. Teleological relations (Giunchiglia and Fumagalli, 2017) are based on *Subject-Verb-Object* triplets, while ontological relations are modeled after common relations found in upper ontologies, such as *is-a*, *part-of*, etc. Previous attempts to apply this annotation model to medical text show that it is possible to obtain near-human-level accuracy with as little as 600 training examples (Piad-Morffis et al., 2019b). It has been previously used in three editions of the *eHealth-KD Challenge* shared annotation campaign [2].

The main objective of this research is to build a manually annotated corpus that can be used to train knowledge discovery systems for analyzing the COVID-19 research. For this purpose, we apply the SAT+R annotation model to English sentences in the CORD-19 corpus, manually annotating a small training set that can bootstrap a text-mining process for the entire corpus. The main contributions of this research are:

- The manual annotation of 500 sentences hand-picked from the CORD-19 corpus with the SAT+R annotation model, using a volunteer crowd-based approach with 2 versions of each annotation made by different non-expert annotators.

- The implementation and evaluation of a baseline text-mining pipeline trained on the manually annotated sentences.

- The application of the text-mining pipeline to the full CORD-19 corpus with an analysis of the most relevant concepts and relations discovered.

- All the relevant data to replicate and continue this research, including the source code and annotated corpus in BRAT Standoff format (Stenetorp et al., 2012), are available online for the research community[3].

The remainder of this paper is organized as follows. Section 2 introduces the SAT+R annotation model and presents the manually annotated corpus, its statistics and quality metrics, and details about the annotation process. Section 3 presents and evaluates a baseline machine learning pipeline to automatically annotate the raw text in the CORD-19 corpus. Section 4 describes the most relevant concepts and relations automatically extracted from the a larger fraction of CORD-19 corpus. Finally, Section 5 discusses the findings and lessons learned during this research and highlights possible lines of development, and Section 6 presents the conclusions.

## 2 Corpus Description

This section introduces the *CORD-ANN* corpus, an ongoing linguistic resource which contains semantically annotated sentences extracted from COVID-related research papers in the *CORD-19* corpus. The corpus is constructed following the SAT+R annotation model (Piad-Morffis et al., 2019a), which is based around *Concepts* linked by *Actions*, and additional ontological and teleological relations, such

[2]https://ehealthkd.github.io

[3]https://github.com/knowledge-learning/cord19-ann

| Metric | Total | A | B |
|---|---|---|---|
| *Sentences* | 1000 | 500 | 500 |
| *Entities* | 10201 | 5110 | 5091 |
| Concept | 8231 | 4154 | 4077 |
| Action | 1868 | 916 | 952 |
| Reference | 102 | 40 | 62 |
| *Relations* | 9444 | 4735 | 4709 |
| in-context | 2239 | 1077 | 1162 |
| has-property | 2185 | 1146 | 1039 |
| target | 1686 | 845 | 841 |
| subject | 1269 | 640 | 629 |
| is-a | 647 | 322 | 325 |
| in-place | 366 | 177 | 189 |
| causes | 297 | 161 | 136 |
| has-part | 217 | 114 | 103 |
| entails | 207 | 88 | 119 |
| in-time | 170 | 87 | 83 |
| same-as | 161 | 78 | 83 |
| *Attributes* | 606 | 329 | 277 |
| uncertain | 228 | 119 | 109 |
| emphasized | 204 | 114 | 90 |
| negated | 142 | 78 | 64 |
| diminished | 32 | 18 | 14 |

Table 1: Summary statistics for the *CORD-ANN* corpus.

as *is-a*, *has-property*, *causes*, *entails*, among others. During an initial annotation trial it was identified that *Predicates*, one of the semantic types defined in SAT+R, produced a large degree of disagreement among annotators while providing little additional information, and for this reason this element is not considered. Figure 1 shows three real sentences annotated with a variety of the semantic elements defined in the SAT+R annotation model.

Table 1 shows the total number of annotated elements. The annotated corpus has 500 sentences manually selected from the *CORD-19* corpus. Each sentence was manually annotated by 2 different annotators who where not allowed to share their annotations. Table 1 shows a fine-grained description of the corpus annotations. A total of 10, 201 entities and 9, 444 relations were annotated, averaging 10.20 entities and 9.44 relations per sentence.

A manual pre-selection process was carried out to choose the sentences that contain the most relevant content. The annotation process is realized in batches of 5 sentences, given the linguistic complexity of the academic language of the corpus and the fact that the annotators are not native English speakers. That being said, the majority of the annotators were undergraduate student or graduate students of different university degrees with at least a B2 English level. The annotation procedure was adapted from the methodology proposed by Piad-Morffis et al. (2019a). The tool used for annotation is BRAT (Stenetorp et al., 2012) given the simplicity of its user interface. Configuration files and related infrastructure are published in the project repository.

The annotators were recruited through social media and most are from the academic institutions to which the authors are affiliated. A total of 21 different annotators were involved in the corpus creation, although several more showed some degree of interest but didn't complete any annotation batch. The degree of involvement varied widely, since two annotators account for approximately half of the corpus (51 and 48 batches respectively) while 9 annotators submitted only one batch. An annotation guide with several examples was published online, and the first batch from every annotator was cross-checked by the authors to provide feedback. Afterwards, a continuous annotation campaign was managed through social media, with regular periods in which the annotators joined in an online forum to ask for clarifications or share their suggestions.

The annotation process was carried out from March 28th until June 9th, when the first 500 sentences were completed. At the moment of writing, the annotation campaign has been temporarily halted in order to analyze the partial results obtained and decide the best course of action for the continuity of this research.

Table 2 shows the agreement score between each pair of annotations for each type of semantic element. The metrics reported are precision, recall and $F_1$ computed as a micro-average between every pair of sentences doubly annotated. Since the $F_1$ metric is symmetric with respect to precision and recall, these are taken with respect to an arbitrary first annotator for each sentence. Overall, the agreement for entities is higher than for relations. The most difficult semantic relations to annotate, in terms of agreement, are *entails*, *causes* and *has-part*, while the easiest are the teleological relations *subject* and *target*, followed by the ontological relations *is-a* and *has-property*.

| Annotation | Recall | Precision | $F_1$ |
|---|---|---|---|
| *Entities* | 0.6367 | 0.6418 | 0.6392 |
| Concept | 0.6699 | 0.6852 | 0.6775 |
| Action | 0.5027 | 0.5424 | 0.5218 |
| Reference | 0.2625 | 0.7000 | 0.3818 |
| *Relations* | 0.4875 | 0.4982 | 0.4928 |
| target | 0.6416 | 0.6381 | 0.6398 |
| subject | 0.6094 | 0.6418 | 0.6252 |
| has-property | 0.5454 | 0.6085 | 0.5752 |
| is-a | 0.5333 | 0.5303 | 0.5318 |
| in-context | 0.4119 | 0.3806 | 0.3956 |
| in-time | 0.3333 | 0.3513 | 0.3421 |
| in-place | 0.2524 | 0.2385 | 0.2452 |
| causes | 0.1264 | 0.1692 | 0.1447 |
| has-part | 0.1142 | 0.1538 | 0.1311 |
| entails | 0.0909 | 0.0789 | 0.0845 |

Table 2: Relative agreement between annotators for each type of annotation.

## 3 Baseline Text-Mining Pipeline

This section presents a simple machine learning pipeline for the automatic annotation of entities and relations in raw sentences from the CORD-19 corpus following the annotation model described in Section 2. This pipeline is trained on the $1,000$ manually annotated sentences (i.e., the two versions of each annotated sentence), and executed on the remaining of the CORD-19 corpus. A high-level overview of the pipeline, shown in Figure 2, is composed of the following steps:

1. Sentences are tokenized and syntactic and morphological features are extracted from each token (using the spaCy[4] library).

2. The annotated entities are converted from BRAT's Standoff format to a BILOUV encoding (i.e., **Begin**, **Inside**, **Last**, **Out**, **Unit** and *oVerlap*).

3. A CRF model $M_E$ is trained on the token features to predict the BILOUV encoding.

4. Each relation pair is converted to a set of aggregated features, and negative relation pairs are randomly sampled.

5. A linear model (logistic regression) $M_R$ is trained on relation pairs to predict the 10 *Relation* classes in Table 1 plus and additional *NONE* relation label.

[4] https://spacy.io

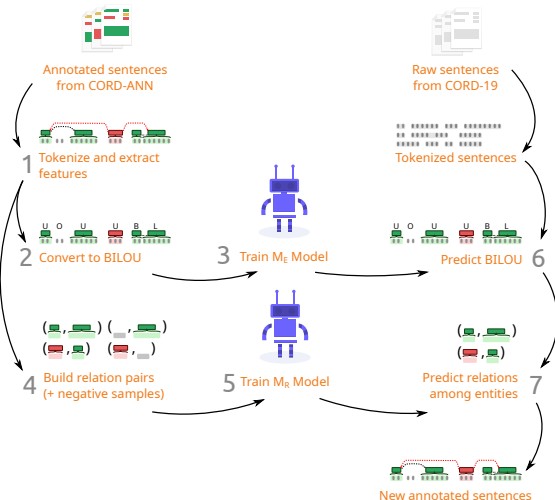

Figure 2: Illustrative representation of the text-mining pipeline designed in this research.

6. The entity model $M_E$ is executed on unlabeled sentences and the result is converted from BILOUV encoding to BRAT's Standoff format.

7. The relation model $M_R$ is executed on the pairs of entities predicted in the previous step.

For the entity model $M_E$, the syntactic and morphological features include lemma, coarse and fine-grained part-of-speech, dependency labels, general-purpose entity labels (e.g., *PERSON*, *LOCATION*, etc.), word shape, and several flags for specific patterns such as emails, numbers, and URLs. For the relation model $M_R$, the aggregated features correspond to the features of the tokens that comprise the two entities that participate in the relation, as well as the features of all the tokens in the smallest sub-tree of the dependency tree that contains both entities.

The ultimate purpose of these models is to automatically extract relevant knowledge from the unlabeled pool of sentences. Taking into account the complexity of this natural language comprehension task, there is always a trade-off between extracting as much knowledge as possible (i.e., maximizing recall) versus extracting knowledge as accurately as possible (i.e., maximizing precision). However, this trade-off can be explicitly controlled by measuring a degree of uncertainty $\sigma$ in the models' predictions, and only outputting the elements (i.e., entities and relations) whose uncertainty is below a given threshold $\sigma^*$. For the entity model $M_E$, the raw marginal probabilities provided by the CRF

| | Entities $M_E$ | | | Relations $M_R$ | | |
|---|---|---|---|---|---|---|
| $\sigma^*$ | Rec. | Prec. | $F_1$ | Rec. | Prec. | $F_1$ |
| 0.25 | 0.000 | 0.200 | 0.000 | 0.000 | 0.000 | 0.000 |
| 0.50 | 0.013 | **0.887** | 0.025 | 0.000 | 0.000 | 0.000 |
| 0.75 | 0.048 | 0.718 | 0.090 | 0.082 | 0.267 | 0.120 |
| 1.00 | 0.086 | 0.585 | 0.150 | 0.138 | **0.695** | 0.220 |
| 1.25 | 0.134 | 0.539 | 0.214 | 0.231 | 0.457 | **0.304** |
| 1.50 | 0.196 | 0.544 | 0.289 | 0.227 | 0.326 | 0.265 |
| 1.75 | 0.300 | 0.552 | 0.388 | 0.232 | 0.270 | 0.248 |
| 2.00 | 0.437 | 0.543 | 0.484 | **0.244** | 0.243 | 0.243 |
| 2.25 | 0.541 | 0.519 | 0.530 | 0.241 | 0.223 | 0.231 |
| 2.50 | 0.585 | 0.506 | 0.543 | 0.238 | 0.212 | 0.224 |
| 2.75 | **0.592** | 0.502 | **0.543** | 0.237 | 0.210 | 0.222 |

Table 3: Results of 30 independent evaluations for the entity model $M_E$ and relation model $M_R$ aggregated for different values of the uncertainty threshold $\sigma^*$.

model are a possible measure of uncertainty, while for the relation model $M_R$ the same role is played by the raw logits provided by the logistic regression model. In both cases, the uncertainty can be estimated by computing the entropy $H$ of the probability distribution of possible labels, given by the formula:

$$H(E) = \sum_{l_i \in L} -P(E = l_i | \theta) \log P(E = l_i | \theta)$$

Where $E$ is an annotation element, $l_i \in L$ are the possible labels, and $\theta$ are the corresponding model parameters (i.e, the CRF transition probabilities or the logistic regression weights).

To evaluate the entity and relation models and estimate the optimal uncertainty threshold value $\sigma^*$, 30 independent executions of a train-test loop with $80\%$ of the training set are performed for different values of the uncertainty threshold. Table 3 presents the mean precision, recall and $F_1$ for entities and relations in each of the threshold values evaluated. The top result for each metric is highlighted.

This analysis is extended to all different annotation elements and summarized in Table 4. The maximum precision, recall and $F_1$ obtained for each annotation element is presented. Each value corresponds to a potentially different uncertainty threshold.

To better understand the trade-off between precision and recall, Figure 3 shows the precision obtained at different uncertainty thresholds for each annotation element. However, since a very high precision can be achieved with an arbitrarily low recall, we only consider annotations for which the

| Annotation | Prec. | Rec. | $F_1$ |
|---|---|---|---|
| *Entities* | | | |
| Action | 0.961 | 0.221 | 0.327 |
| Concept | 0.887 | 0.684 | 0.576 |
| Reference | 0.000 | 0.000 | 0.000 |
| *Relations* | | | |
| causes | 0.133 | 0.092 | 0.104 |
| entails | 0.008 | 0.005 | 0.006 |
| has-part | 0.112 | 0.040 | 0.052 |
| has-property | 0.572 | 0.282 | 0.269 |
| in-context | 0.716 | 0.352 | 0.399 |
| in-place | 0.320 | 0.131 | 0.171 |
| in-time | 0.254 | 0.077 | 0.105 |
| is-a | 0.371 | 0.282 | 0.296 |
| subject | 0.268 | 0.373 | 0.305 |
| target | 0.330 | 0.460 | 0.366 |

Table 4: Maximum precision, recall and $F_1$ obtained for different annotation elements. Each value was obtained for a potentially different uncertainty threshold.

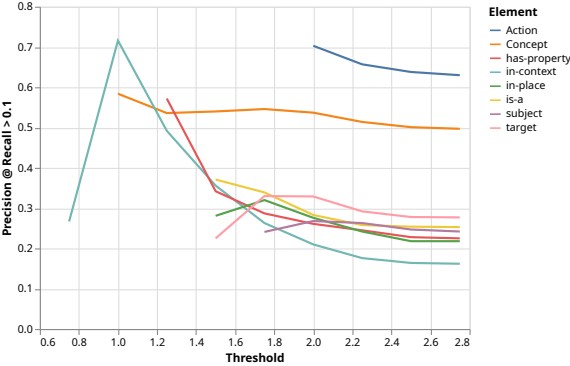

Figure 3: Maximum precision achieved for different semantic elements given a specific uncertainty threshold. Only results with recall above 0.1 are considered.

average recall is above 0.1 at the given threshold level. This guarantees that at least a $10\%$ of the potential number of those semantic elements would be extracted from the unlabeled collection.

## 4   Preliminary Insights in Knowledge Discovery

This section presents a qualitative analysis of a knowledge discovery process that can be performed using machine learning models trained on this type of annotated data. For this purpose, the remaining sentences of the CORD-19 corpus that were not used during the annotation process were fed to the machine learning models $M_E$ and $M_R$ and all pre-

| Type | Instances |
|------|-----------|
| *Sentences* | 100,959 |
| *Entities* | 782,141 |
| `Concept` | 737,838 |
| `Action` | 44,289 |
| `Reference` | 14 |
| *Relations* | 783,534 |
| `has-property` | 360,986 |
| `in-context` | 267,333 |
| `target` | 72,730 |
| `is-a` | 26,070 |
| `subject` | 22,513 |
| `same-as` | 15,520 |
| `in-place` | 12,824 |
| `entails` | 2,058 |
| `causes` | 1,805 |
| `in-time` | 895 |
| `has-part` | 800 |

Table 5: Total number of instances extracted from the unlabeled sentences.

| Concept | Instances | Concept | Instances |
|---------|-----------|---------|-----------|
| *cases* | 5,742 | *COVID-19* | 3,007 |
| *number* | 4,775 | *infection* | 3,006 |
| *patients* | 4,577 | *cell* | 2,411 |
| *SARS* | 4,284 | *epidemic* | 2,315 |
| *model* | 3,755 | *proteins* | 2,224 |
| *data* | 3,686 | *human* | 2,223 |
| *protein* | 3,382 | *RNA* | 2,169 |
| *virus* | 3,222 | *CoV* | 2,168 |
| *granted* | 3,147 | *infected* | 2,153 |
| *cells* | 3,015 | *China* | 2,132 |

Table 6: Most common entities extracted from the *CORD-19* corpus.

| Relation | Source | Destination | Count |
|----------|--------|-------------|-------|
| `is-a` | *SARS* | *coronavirus* | 85 |
| `is-a` | *MERS* | *CoV* | 76 |
| `is-a` | *SEIR* | *model* | 64 |
| `is-a` | *influenza* | *virus* | 61 |
| `is-a` | *A549* | *cells* | 46 |
| `has-property` | *cases* | *confirmed* | 439 |
| `has-property` | *patients* | *severe* | 357 |
| `has-property` | *cases* | *severe* | 273 |
| `has-property` | *number* | *basic* | 245 |
| `has-property` | *cases* | *imported* | 213 |

Table 7: Most common instances for the relations *is-a* and *has-property*.

dicted entities and relations were stored. Table 5 presents the total number of sentences processed as well as entities and relations extracted. Given the relatively low performance of the machine learning models, a simple post-processing was introduced to remove the most obvious sources of errors, such as numbers and mathematical symbols, that were incorrectly detected as entities. As expected, the distribution of extracted elements closely follows the distribution of annotations in the training set (see Table 1).

The 20 most common entities extracted are summarized in Table 6. Unsurprisingly, they correspond to common concepts in the medical literature related to epidemics, treatments, biological entities, as well as some COVID-specific entities and locations. Similarly, Table 7 shows the most common instances of the relations *is-a* and *has-property*, which are the most basic ontological relations. As expected, they correspond mostly to known relations in the medical domain and specifically in the COVID-related literature.

Finally, Figure 4 shows a cherry-picked graph of relations built around the concept *COVID-19*. This graph was constructed by sampling the most common relations that involve this concept, manually eliminating irrelevant tuples, such as *in-context*, and compacting very similar relations (in terms of lemma) into the same nodes. The graph shows interesting relations, such as known symptoms (grouped under the *causes* relation) and a number of properties that are reported among the biomedical literature in the *CORD-19* corpus.

## 5  Discussion and Future Work

This section discusses two important insights that arise from this ongoing research. First, we analyze the quantity and quality of the extracted knowledge, in an attempt to validate the approach and estimate the impact of its components. Second, we discuss some lessons learned during the annotation process in the hope that future research can further improve on our work.

Arguably, two of the most relevant annotation patterns for the purpose of knowledge discovery are ontological relations such as *is-a* and *has-property*, and the Subject-Action-Target triplets via the *target* and *subject* relations. The fact that these 4 relations have a relatively high number of instances extracted is promising. In contrast, the relations *en-*

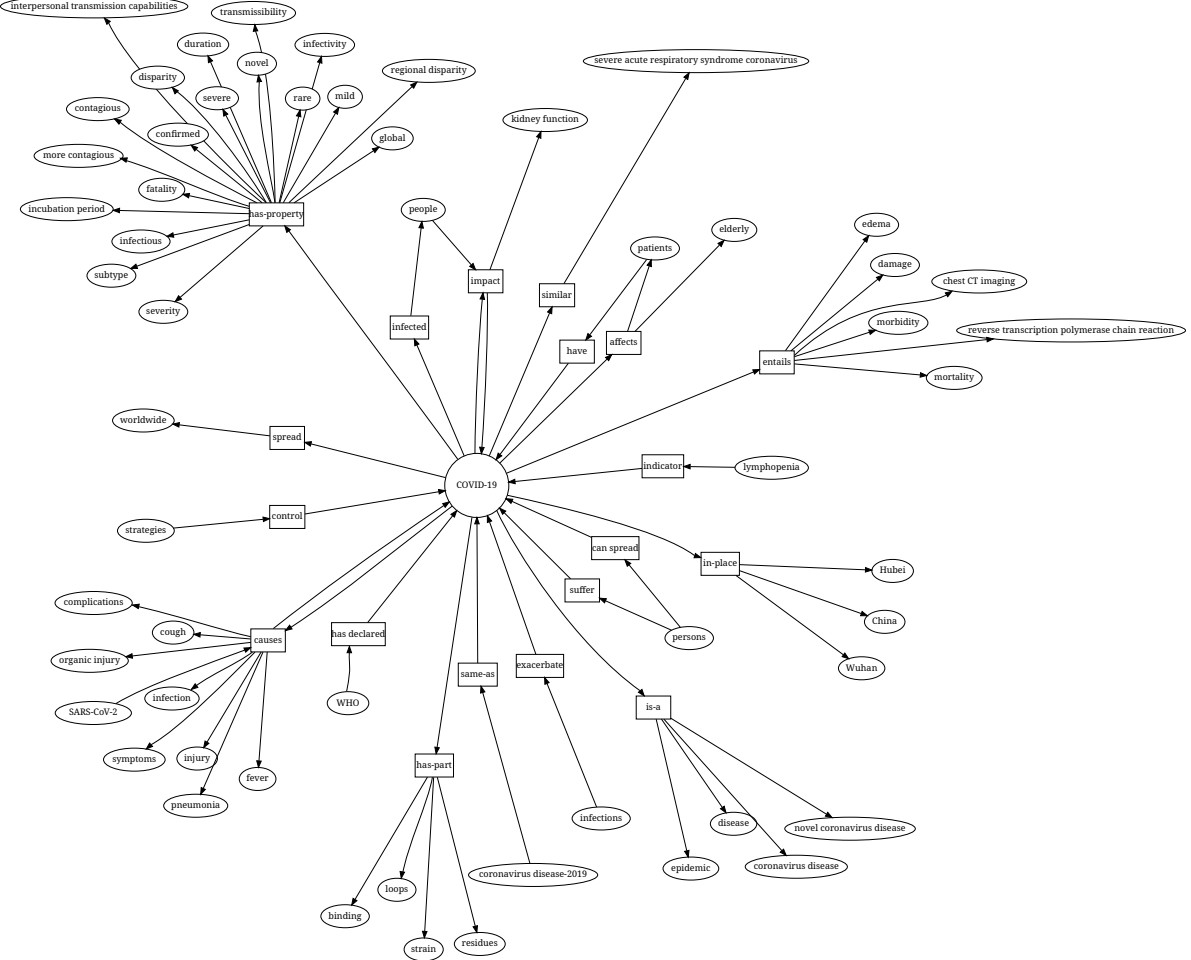

Figure 4: Semantic graph of the most relevant concepts related with the entity *COVID-19*.

*tails* and *causes* show a significantly lower number of instances, as well as a lower performance. These relations are important from the point of view of knowledge discovery since they could directly link symptoms and treatments with evidences. However, more work is necessary to achieve a reasonable level of performance in their extraction.

The machine learning models presented in this work where designed as an initial baseline. In a practical scenario more powerful models would be deployed. Possible strategies include using a separate CRF model for each entity class, and using a hierarchical model for the relation extraction step. Another possible strategy is to repurpose existing models deployed in similar tasks. According to the 2020 edition of the *eHealth-KD Challenge*, the top performing deep learning models in this task achieve $F_1$ values of 0.82 and 0.63 in entity and relation extraction, respectively (Piad-Morffis et al., 2020). Since they are also built on the SAT+R annotation model, their adaptation to the corpus used in this research is straightforward. However,

it should be considered that the reduced number of training examples in our corpus presents a significant challenge for any machine learning model, which motivates the use of transfer learning techniques. In this sense, additional annotated corpora with SAT+R, such as the one presented in Piad-Morffis et al. (2020) can be used to bootstrap such as system.

Moving forward towards a fully-fledged ontology learning process still requires a significant effort after the annotation and model training. Particularly, since each entity can be potentially represented in the text in different forms, a normalization step would be necessary to group all mentions of similar entities under a single concept. At the moment of writing we are developing approaches for automatic normalization of entity mentions based on *Wikidata*, but the results are still not available. Furthermore, in this work we have not taken into account the prediction confidence beyond its use as a threshold. Weighting the number of mentions of each entity and relation with their respective

confidence could provide an additional metric to determine what to include in the knowledge graph.

One surprising and positive conclusion of our research is that crowd-based annotation efforts like this one, even with complex cognitive tasks involving deep semantics, are feasible. The annotators that participated voluntarily in this research were motivated by purely altruistic reasons, since there was no monetary incentive. We argue that, even if the COVID-19 pandemic played a large role in this motivation, in general people can be motivated for this kind of work because of the positive social impact of the research. Regarding annotation quality, there is a non-trivial amount of initial tutoring and feedback necessary, but once an annotator acquires a certain level of expertise, these efforts begin to pay-off since the experienced annotator becomes a potential tutor for new recruits.

To improve the annotation process, two considerations are possible. First, the annotators can be automatically evaluated on a small trial set with automatic feedback. This way, a minimum level of initial expertise is guaranteed. Afterwards, it is interesting to apply active learning strategies (Settles, 2009) to automatically select which sentences to annotate. In addition to standard active learning approaches, where a classifier-based uncertainty or informativeness measure is used, in this context the inter-annotator agreement could be directly used to re-sample sentences for which the agreement is low, so that the more complex sentences receive more annotations.

## 6    Conclusion

This paper presents the preliminary results of ongoing research. The majority of current research in the CORD-19 corpus uses unsupervised or semi-supervised approaches for knowledge discovery. We propose a supervised approach for extracting semantic relations and concepts in scientific articles using the CORD-19 corpus. For this purpose we annotated $500$ sentences using a general purpose semantic model. We propose a baseline text-mining pipeline, trained on this data and executed on $100,959$ additional sentences of the CORD-19 corpus, for automatically extracting relevant knowledge. This approach allows the discovery of relevant facts mentioned in research papers with fine-grained semantics, including causality, compositionality, and contextual dependencies. The annotated corpus and baseline implementation can be used as a starting point for developing more powerful knowledge discovery systems that can automatically analyze the growing body of scientific research related to the COVID-19 epidemic and similar future scenarios.

## Acknowledgments

**Funding:**   This research has been supported by a Carolina Foundation grant in agreement with University of Alicante and University of Havana.     Moreover, it has also been partially funded by both aforementioned universities, the Generalitat Valenciana (*Conselleria d'Educació, Investigació, Cultura i Esport*) and the Spanish Government through the projects LIVING-LANG (`RTI2018-094653-B-C22`) and SIIA (`PROMETEO/2018/089`).

This research was possible thanks to the voluntary effort of several annotators, including: Abel Molina Sánchez, Alejandro Klever Clemente, Daniel Alejandro Cárdenas Cabrera, Daniel Valdés Pérez, Gabriela Bárbara Martínez Giraldo, Gabriela Fernández Méndez, Houcemeddine Turki, Inti Antonio Blanco González, We sincerely thank all of the above and the remaining anonymous annotators. Finally, we would like to thank the anonymous reviewers. Their helpful comments have been considered in the current version of this paper.

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
