# OpenReview forum: "Knowledge Discovery in COVID-19 Research Literature"
_EMNLP/2020/Workshop/NLP-COVID — NLP-COVID19-EMNLP Poster_

### Official Review · AnonReviewer3 · 2020-09-15
**Reasonable extension of an existing approach to Covid-19 - but is the existing approach really valuable?**

**Rating:** 6
**Confidence:** 1

**Review:**

This paper presents an approach to "knowledge discovery" on the Covid-19 literature, based on previous work in the more general biomedical domain. Its main contribution is a collection of annotated sentences, annotated by volunteers recruited via social media, for use as test and training data, complete with inter-annotator agreement scores. It also includes a baseline system trained on the annotated sentences, an analysis of the corpus, and the source code and data are available online.

The paper builds upon a body of work using the SAT+R (Subject-Action-Target plus Relations) formalism, a general purpose semantic model that has been applied to the biomedical domain, including its deployment as a standard in shared tasks.

Reasons to accept: This seems like a reasonably competent application of an existing research programme to Covid-19. It is always good to see datasets and evaluation resources. Their crowdsourced annotation is interesting - but the success may be dependent on extra goodwill being available in the time of Covid-19, so it's not clear how it will transfer.

Reasons to reject: The value of SAT+R annotations is unclear to me. The inter-annotator F scores they have achieved are not great (.64 for entities and .49 for relations, with some relations having agreement as low as .08) - this may reflect the intrinsic subjectivity of the task, or a lack of developement of guidelines, or may be a result of their crowdsourced annotation approach. The F scores for their baseline system are very low - 0.58 - 0.00 for entities and 0.40 - 0.01 for relations - this is not a viable baseline, and is purely of value for comparisons. The authors do state that "Previous attempts to apply this annotation model to medical text show that it is possible to obtain near-human-level accuracy with as little as 600 training examples", but even if F scores comparable to the inter-annotator level are possible, those scores are not especially high. Given that the application for these Subjects, Actions, Targets and Relations is to build graphs, the probability of errors in even a small graph becomes very high. It may be that these efforts are part of a long-term research program which will reach the appropriate levels of accuracy by incremental progress in the fullness of time - if so, I question its application to the Covid-19 domain, where applications ready for use by biomedical researchers etc. are needed urgently.

Conclusion: The work looks like a competent extension of an already-existing research programme to the Covid-19 domain, and the evaluation resources it has produced may be of use to the knowledge discovery subcommunity. I am not at all familiar with these kinds of knowledge discovery approaches so I find it hard to gauge the importance of the research or its likelihood of producing outcomes of value to biomedical researchers - especially of value to Covid-19 researchers in a timely enough manner. However, given that other people see fit to publish this kind of work, I very tentatively recommend publication.

---

### Official Review · AnonReviewer1 · 2020-09-20
**An improvable pipeline for COVID-19 related knowledge discovery**

**Rating:** 6
**Confidence:** 2

**Review:**


The authors label a part of a covid-19 research literature dataset and train a ML pipeline for annotating covid-19 research. They apply the developed model, and with simple text analysis and qualitative interpretation they create an ontology of covid-19.

Reasons to accept:
- The authors use a large group of humans (21) to annotate data
- They follow a simple straightforward modeling process (shown in Fig.2) to label total corpus.
- The pipeline, although takes a small labeled dataset provides promising results (see Fig.4 covid-19 ontology)
- The paper is well-written, and provides in detail the scope of the pipeline, how they labeled the data, and how they performed model deployment and the evaluation of results.

Reasons to reject:
- The reported Precision-Recall is in some cases really low (e.g. has-part, table-4). I suspect this is due to the relatively small training-dataset (500 sentences).
- The ME, and ML models are CRF and a logistic regression. The authors did not try different model architectures and do not report the predictive ability of other baseline models (e.g. naive bayes) in order to appreciate CRFs and logistic regression's performance.
- Because of pipeline's low performance, they have to perform a qualitative analysis to faciliate COVID-19 knowledge discovery

I appreciate the authors efforts. Therefore I kindly encourage them to address (if possible) following issues:

- provide the accuracy of other models in place of ME (CRF) and ML (LL) and report your accuracy.
- provide the predictive ability of standard models for annotating data in other cases in order to provide a measure for your models' accuracy. (What is a general acceptable precision and recall for tagging e.g. has-part?)
- Since considerable time has passed since the submission of the paper. Have you improved the pipeline? Did you label more data until now?
- Since data are available in Kaggle, and many researchers have analyzed this data, are there other models by researchers that performed the same task? If yes, please compare your pipeline with theirs, in order to show why is yours is better.

---

### Official Review · AnonReviewer2 · 2020-09-24
**Use-case of Information Extraction**

**Rating:** 6
**Confidence:** 4

**Review:**

This paper is a study of a use-case on using an ontology of information extraction to create a knowledge graph around Covid-19 scientific literature.
It is hard to get information extraction "right". Those frameworks that allow for high performance (F1) are often too simplistic, and those that are too fine-grained do not obtain a performance level which is useful. This is a nice example of a middle-ground: a rather fine-grained extraction of relationships, with detailed performance analysis. While it remains low, the analysis shows how to tune recall vs precision.

[strengths]
 * a good example on how to leverage existing research frameworks (in inf extraction in this case) for a new use-case
 * an annotated dataset which can be helpful for future research. QUESTION: will this dataset be released? (I found the *.ann files on your github: are those all the annotations?)
 * a complete study: annotations, algorithm, performance analysis, first results

[weakness]
 * The final outcome is obtained by gluing a number of parts together. While this is very pragmatic, it raises a number of questions (see below). This is, if the end goal is to obtain a general graph, then a from-scratch approach might look different from what is proposed. Of course, a from-scratch approach is not what this workshop is looking for
 * the ML algorithms used are probably the weakest part of this proposal. The weight these days is to justify non-neural methods, and no ablation study of the features used is mentioned. The current pipeline approach consists in (i) extracting entities, (ii) extracting relations, (iii) constructing the knowledge graph. Many things could slip through the cracks between all those steps, but the magnitude of this is unknown, and despite the existence of joint extraction models they are not mentioned.


Other questions:
        * Is the annotation of relations done wrt the original gold annotations, or with respect to the result of the entity-extraction step?
	* “selected by the authors from the CORD-19 corpus”. What does this mean? Did you interact with the authors of that corpus?
	* What are the output classes of M_R? Are those all 10 relations of Table 2 + None? Following your pipeline approach, did you try with first predicting if there is a relation, and then which relation it is?
	* The BILOUV encoding is redundant I believe. Wouldn't you obtain better performance by predicting a simpler encoding?
	* Fig 3 shows that the threshold obtaining the best trade-off is different for each entity type. This might indicate that independent sequence prediction models could obtain better performance, in particular as a conflict in the extraction of entities (eg, one span annotated as Concept and also as Action) is not really an issue for your final goal of displaying a graph of knowledge.
	* How does the number of supporting documents enter in your knowledge graph? This is, how do you combine the confidence of a prediction of your model, with the number of times that prediction was made (eg, <lymphopenia, indicator, Covid-19>  predicted only once with high confidence vs predicted often with medium confidence)
	* Instead of Fig 3, a prec-recall curve (or better, a prec-recall-gain curve [1]) might be more indicative of the efficiency of your
algorithms

[1] Flach, Peter, and Meelis Kull. "Precision-recall-gain curves: PR analysis done right." Advances in neural information processing systems. 2015.